# The Identification of Ethidium Bromide-Degrading Bacteria from Laboratory Gel Electrophoresis Waste

**DOI:** 10.3390/biotech11010004

**Published:** 2022-02-24

**Authors:** Vikram Pal Gandhi, Kavindra Kumar Kesari, Anil Kumar

**Affiliations:** 1Vector Biology and Control Division, Rajendra Memorial Research Institute of Medical Sciences, Agamkuan, Patna 800007, India; vikramk2991@gmail.com; 2Department of Bioproducts and Biosystems, School of Chemical Engineering, Aalto University, 00076 Espoo, Finland; 3Department of Life Sciences, School of Natural Sciences, Central University of Jharkhand, Cheri-Manatu, Kamre, Kanke, Ranchi 835222, India

**Keywords:** bioaccumulation, risk assessment, inhibition zone, biotransformation, 16S-rRNA, *Proteus terrae*, *Morganella morganii*, pathogenicity

## Abstract

Ethidium bromide (EtBr) is widely used in most laboratories to detect nucleic acids in gel electrophoresis applications. It is a well-known carcinogenic and mutagenic agent, which can affect biotic components of the place in which it is disposed. Usually the gel-waste is either buried in the ground or incinerated, whereas the liquid waste is disposed of down the sink following the recommended methods of treatment. The recommended methods do not involve biological potential, but rather make use of chemicals, which may further deteriorate soil and water quality. The present study identifies and characterizes the EtBr-degrading bacterial isolates BR3 and BR4. A bibliographic review of the risk status of using these isolates for the treatment of lab waste in laboratory settings is also presented. BR3 was identified as *Proteus terrae* N5/687 (LN680103) and BR4 as *Morganella morganii* subsp. *morganii* ATCC 25830 (AJ301681) with 99.9% and 99.48% similarity, respectively, using an EzBioCloud microbial identifier. The literature revealed the bacterium *Proteus terrae* as a non-pathogenic and natural microflora of humans, but *Morganella morganii* as an opportunistic pathogen. These organisms belong to risk group II. Screening the sensitivity of these isolates to antibiotics revealed a sufficient number of antibiotics, which can be used to control them, if required. BR3 and BR4 exhibited resistance to individual antibiotics, ampicillin and vancomycin, whereas only BR3 was resistant to tetracycline. The current investigation, along with earlier reported work on these isolates, identifies BR3 as a useful isolate in the industrial application for the degradation of EtBr. Identical and related microorganisms, which are available in the culture collection repositories, can also be explored for such potential to formulate a microbial consortium for the bioremediation of ethidium bromide prior to its disposal.

## 1. Introduction

Nowadays, molecular biology has become a routine field of research in which ethidium bromide is used as a staining dye for nucleic acid visualization. Ethidium bromide (EtBr) intercalates between adjacent nitrogenous bases of nucleic acids, enhancing the fluorescence of nucleic acid molecules under UV light [1,2,3].

EtBr containing wastes are recommended to be decontaminated or treated before they are disposed. There are a number of ways these wastes can be treated or decontaminated, including treatment with bleach, incineration, processing of the solution through Rohm and Haas Amberlite XAD-16 resin, Fenton-like reaction using MNCs (magnetic nanocatalysts) and other methods and products [4,5,6,7,8,9]. However, some of these methods used agents, such as sodium nitrite and hypophosphorous acids, which are still harmful [4,10].

In addition to these traditional methods, bioremediation has been considered as an alternate method for detoxification of these xenobiotic compounds. There are different research groups working on EtBr biodegradation and have identified some EtBr-resistant and EtBr-degrading microbes, including *Aeromonas hydrophila, Bacillus*
*species, B. thuringiensis, Neisseria canis, N. subflava, N. macacae, Pseudomonas chlororaphis* and *P. putida* from uncontaminated soil [10,11,12]. Two unidentified bacteria, BR3 and BR4, have also been reported from EtBr containing lab waste (agarose gel), which exhibited EtBr degradation and bioaccumulation, respectively [13]. A plan of a lab model for EtBr degradation using BR3 and BR4 was also proposed, which has to be explored and evaluated further [13]. However, aseptic operation and containment is one of the important aspects of laboratory-based or industrial applications of any organism or bioreactor, and hence determining the pathogenicity and control measures of pathogens becomes the important criteria for such applications or operations of a bioreactor [14]. There are different virulence-determining factors, including fimbriae, flagella, urease, IgA proteases, amino acid deaminases, invasiveness, hemolysins, capsular polysaccharide and LPS, which contribute to the pathogenicity of a bacterium depending upon its genus [15,16].

Thus, either testing all these parameters or deciphering the identity of these isolates and comparing them to the known pathogenicity of related organism helps to assign risks, while working with these isolates for the treatment of lab waste at laboratory settings or at an industrial scale. Hence, the objectives of the present study are to elaborate on the characterization of these bacterial isolates for their application and perspectives on safety reported earlier, including their antibiogram, identification, phylogenetic relationship and pathogenicity, which is helpful to the planning strategy of further evaluations of EtBr biodegradation.

## 2. Materials and Methods

The experimental work plan of the study is outlined in Figure 1.

### 2.1. Bacterial Isolates and Culture

The bacterial isolates BR3 and BR4 were procured from a laboratory at the Department of Life Sciences, Central University of Jharkhand, Brambe, Ranchi-835205, Jharkhand, India [13]. These isolates were characterized as pure cultures and were being maintained at 4 °C, separately, on a nutrient agar plate as well as a nutrient agar plate supplemented with EtBr. Their glycerol stocks were also maintained in our lab. These bacteria, respectively, were reported to bio-transform and bioaccumulate EtBr [13]. However, no toxic impact of EtBr accumulation or EtBr biotransformation was reported [13].

### 2.2. Antibiotic Sensitivity Profiling

The bacterial isolates BR3 and BR4 were screened to check their resistance/sensitivity to ampicillin, a commonly used resistance marker in cloning experiments. Bacterial isolates were grown in LB broth overnight (approx. 17 h) at 37 °C in a shaking incubator at 160 rpm. These cultures of bacterial isolates were spread, with a sterile cotton swab, onto LB agar plates with and without ampicillin (100 µg/mL). These plates were incubated overnight for up to 24 h on LB agar plates at 37 °C. The growth of bacterial isolates was recorded to measure their resistance to antibiotics. Resistance to other antibiotics was tested by a standard method, single disc diffusion on Mueller–Hinton agar, using an Antibiotic Sensitivity Teaching kit HTM002-15PR from HiMedia Laboratories Pvt. Ltd., Mumbai, India) [17]. Briefly, the cultures were grown overnight (approx. 17 h) in culture broth (LB) at 37 °C in a shaking incubator at 160 rpm. Cells were collected by centrifugation (5000 rpm, 2 min, 29 °C) and the pellet was rinsed with normal saline. Cells were suspended in sterile normal saline. Bacteria were spread on separate Mueller–Hinton agar plates using a sterile cotton swab. Discs of different antibiotics, such as chloramphenicol (30 µg/disc), vancomycin (30 µg/disc), kanamycin (30 µg/disc), gentamicin (10 µg/disc), and tetracycline (30 µg/disc), were placed on inoculated Mueller–Hinton agar plates. These inoculated plates were incubated at 37 °C for 24 h. The diameter of the inhibition zones were recorded in millimeters (mm). Interpretations of the isolates were made as resistant (R), intermediate (I) and susceptible (S), according to the reference range of the test kit manual and Wayne [18].

### 2.3. Gel Electrophoresis of Crude Bicterial Lysate and Plasmid Preparation

The preparation of bacterial cell lysate and gel electrophoresis was performed following the principles presented in a laboratory manual authored by Sambrook and Russell [19]. Briefly, the bacteria were grown in LB broth: (a.) without EtBr and (b.) with 30 μg/mL EtBr and 100 μg/mL ampicillin. Cell pellets of overnight grown bacterial cultures were washed and re-suspended in 567 μL TE buffer (10 mM Tris, 1 mM EDTA, pH 7.4). SDS (30 μL of 10% SDS) and proteinase K (3 μL of 20 mg/mL proteinase K) were added and mixed well, followed by incubation for an hour at 37 °C for cell lysis. The lysates of respective cultures were divided into two sets. One set of lysates was treated with RNase for 30 min at 37 °C. An aliquot of 10 µL of each cell lysate were mixed with 6X gel loading dye (6X: glycerol 30% *v*/*v*, bromophenol blue 0.25% *w*/*v*, xylene cynol 0.25% *w*/*v*) and loaded into wells followed by agarose gel (1%) electrophoresis (5 V/cm). The positions of the nucleic acid bands in the gel were visualized under trans-UV after staining the gel with ethidium bromide (0.5 μg/mL). An image was taken in the AlphaImager MINI gel documentation system (proteinsimple, San Jose, CA, USA). Presence of plasmid was confirmed by plasmid extraction using PureLink Quick Plasmid Miniprep Kit following the procedure given in kit’s manual.

### 2.4. 16-rRNA Gene Based Phylogenetic Analysis of Bacterial Isolates

The phylogenetic analyses of bacterial isolates BR3 and BR4 were performed using the 16S-rRNA gene sequence [20]. The bacterial isolates BR3 and BR4 were subjected to 16S-rRNA PCR and sequencing was performed as per Gandhi and Kumar [21]. The 16S-rRNA gene specific ~1.5 kb amplicons from BR3 and BR4 isolates were sequenced by an automated genetic analyzer Applied Biosystems 3500xL (Applied Biosystems, Foster City, CA, USA) by Banaras Lab, India. The chromatogram quality of the nucleotide sequences were checked by Finch TV (Geospiza Inc., Seattle, WA, USA) chromatogram viewer based on its default threshold value. Each nucleotide peak was also checked manually for correct nucleotide bases in the sequences. Quality checked nucleotide contigs were aligned using the BioEdit program [22], CAP:3 (contig alignment program) [23]. The aligned 16S-rRNA gene-specific bacterial nucleotide sequences were further used for identification through the EzBioCloud microbial identifier (http://www.ezbiocloud.net/identify (accessed on 22 October 2018)) [24]. The nucleotide sequence specific to the 16S-rRNA gene of isolates was uploaded in the identification module of EzBioCloud (CJ Bioscience, Inc., Seoul, Korea) and the isolates with their respective type strains were identified on the basis of maximum similarity. The FASTA file of the top 10–20 most similar aligned sequences was downloaded and used for phylogenetic analysis using the MEGA-X (Institute of Molecular Evolutionary Genetics, State College, PA, USA) offline tool [25]. A phylogenetic tree was constructed in the MEGA-X offline tool using the neighbor-joining method and 500 bootstraps. The 16S-rRNA nucleotide sequences of isolates BR3 and BR4 were submitted to the NCBI database using the online submission tool Banklt (NLM, Bethesda, MD, USA).

### 2.5. Pathogenic Status of BR3 and BR4

The pathogenic status of BR3 and BR4 was determined on the basis of their identification and pathogenicity status report of closely related genus and species.

## 3. Results and Discussion

### 3.1. Antibiotic Resistance Assay and Gel Electrophoresis of Crude Lysate

The tested bacterial isolates revealed different susceptibilities to the tested antibiotics. Both isolates, BR3 and BR4, were resistant to ampicillin and vancomycin, but sensitive to gentamicin and kanamycin (Figure 2 and Figure 3, Table 1). BR3 was resistant to tetracycline but sensitive to chloramphenicol, whereas BR4 responded in the opposite manner as sensitive to tetracycline but intermediate to chloramphenicol (Table 1; Figure 3). No study reported the antibiotic profile of *P. terrae* (BR3) *M. morganii* (BR4) from laboratory gel electrophoresis waste. Generally, this has been reported from post-operative wound and urinary tract infections [26].

The gel electrophoresis of the crude lysate revealed the presence of a fluorescent band observed below the genomic/chromosomal DNA, but significantly above the usual position of the RNA band. The gel electrophoresis of the plasmid preparation using the kit confirmed the presence of plasmid band observed just above the molecular weight marker band of 15,000 bp (Figure 4). This plasmid probably bears the genes responsible for its resistance to antibiotics; however, it was not confirmed by plasmid curing as determined by Patil and Berde [12] in their studies. This plasmid may not be attributed to bear genes for EtBr degradation, as it was already revealed by a plasmid curing study that the EtBr-degrading trait is localized on chromosomal DNA [12].

### 3.2. Identification and Phylogenetic Status of Bacterial Isolates

The 16S-rRNA gene sequence was used for the identification of bacterial isolates and phylogenetic analysis. Nucleotide sequencing of the 16S-rRNA gene of isolates BR3 and BR4 resulted in 1034 and 963 nucleotide long sequences, respectively. The respective nucleotide sequences of BR3 and BR4 were searched in the EzBioCloud microbial identifier, which revealed a 99.90% and 99.48% identity match to a *P. terrae* N5/687 (LN680103)-type strain (T) and *Morganella morganii subsp. morganii* ATCC 25830 (AJ301681) (T), respectively.

The EzBioCloud microbial identifier was used for a 16S-rRNA gene sequence similarity search of bacterial isolates with type strain (T) microbial cultures. The phylogenetic trees of respective sequences are shown in Figure 5 and Figure 6. The phylogenetic tree shows a close clustering of BR3 with *P. terrae*, followed by *P. terrae* N5/687(T) and *P. cibarius* JS9 (T) (Figure 5) [27]. BR4 clustered with *M. morganii subsp. morganii* followed by *M. morganii subsp. morganii* ATCC 25830 (T) and *M. morganii subsp. sibonii* DSM 14850 (T) in its phylogenetic tree (Figure 6). The nucleotide sequences of BR3 and BR4 were submitted to GenBank under the accession numbers KY684830.1 and KY697117.1, respectively. Thus, the phylogenetic analysis is well supported by our earlier findings, which characterized these isolates as two distinct groups with quite distinct biochemical features, as well as quite distinct mechanisms for managing EtBr in its surroundings [13].

### 3.3. Pathogenicity Status of the Isolates

Isolates BR3 and BR4, which are identified as *P. terrae* and *M. morganii*, respectively, may be described as opportunistic pathogens and may be assigned to risk group II (moderate individual risk, limited community risk, and includes opportunistic pathogens) [28] in the light of earlier studies, where *Proteus* and *Morganella*, both bacterial genera, were revealed to cause skin wounds and urinary tract infections [15,16,18,29]. Additionally, *M. morganii* is an unusual opportunistic pathogen and results in a high mortality rate due to its virulence and increasing drug resistance [15,29], which corroborated our study that BR4 was found in this study to be resistant to the antibiotics ampicillin and vancomycin, thus adding to the pathogenic status of BR4. Out of seven of the reported species of *Proteus,* three species, *myxofaciens*, *terrae*, and *cibarius*, have no report of pathogenicity for humans [16,30,31]. Thus, the clustering of BR3 with *P. terrae* N5/687(T) revealed that isolate BR3 is a non-pathogenic bacterium. Although BR3 is resistant to some tested antibiotics, such as ampicillin, vancomycin, and tetracycline, it is sensitive to other tested antibiotics, such as kanamycin, gentamicin, and chloramphenicol. Thus, BR3 may be characterized as either non-pathogenic to humans or as an opportunistic pathogen causing mild infections of the skin or urinary tract. High EtBr bio-degradation efficiency [13], sensitivity to readily available antimicrobials and its mild severity or non-pathogenicity to humans [16,30,31], makes BR3 a suitable isolate for its use in industrial applications, when the utmost care is taken. On the other hand, the authors discourage the application of BR4 in EtBr biodegradation on an industrial scale, as it is an inefficient bio-degrader [13] with a high pathogenic status and increasing drug-resistant report.

## 4. Conclusions

The present study is a continuation of our previous findings, where the bacterial isolate BR3 is identified as *P. terrae* and BR4 is identified as *M. morganii* subsp. *morganii*. BR3 is capable of biotransforming EtBr efficiently, which is also investigated in this study. By considering these isolates as non-pathogenic to humans and its susceptibility to readily available antibiotics, this study did not find potential risks, and hence BR3 (*P. terrae*) may be proposed for use in the bioremediation of EtBr in laboratory settings, when the utmost is taken. This is the first study to report the antibiotic profiles of BR3 and BR4 from laboratory gel electrophoresis waste. Therefore, antibiotic profiling, in the present study, is useful to measure the functional evaluation of the model designed earlier for biodegradation. This application can also be extended to an industrial scale following the standards of industrial scale-up processes.

## Figures and Tables

**Figure 1 biotech-11-00004-f001:**
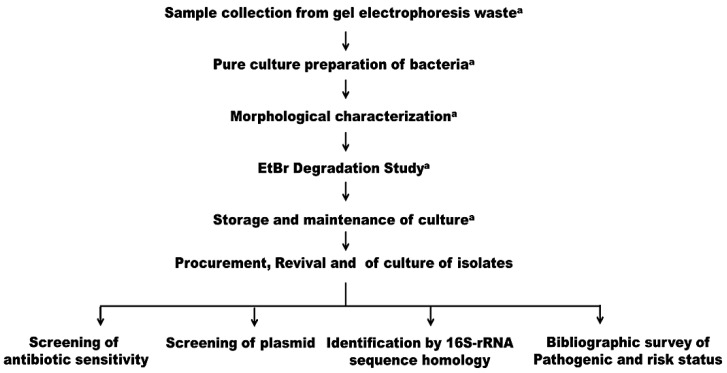
An outline of the work performed with respect to BR3 and BR4. ^a^ represents the work reported earlier [13].

**Figure 2 biotech-11-00004-f002:**
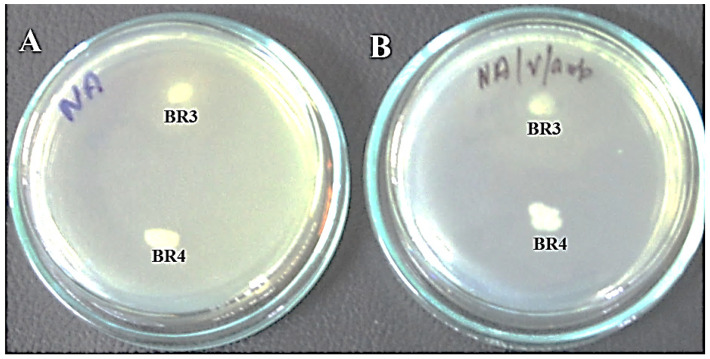
Growth of isolates BR3 and BR4 on nutrient agar plates. (**A**) Nutrient agar plate without ampicillin. (**B**) Nutrient agar plate with ampicillin.

**Figure 3 biotech-11-00004-f003:**
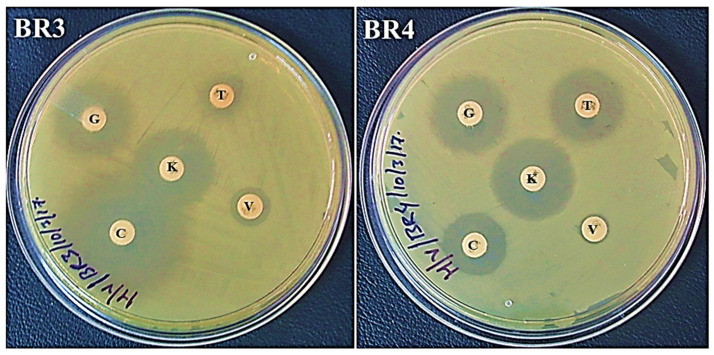
Antibiotic sensitivity profile of BR3 and BR4 on Mueller–Hinton agar plates. Letters on the discs represent antibiotics: C: chloramphenicol (30 µg/disc), V: vancomycin (30 µg/disc), T: tetracycline (30 µg/disc), K: kanamycin (30 µg/disc), and G: gentamicin (10 µg/disc). The inner diameter of the medium holding plate measures 86 mm.

**Figure 4 biotech-11-00004-f004:**
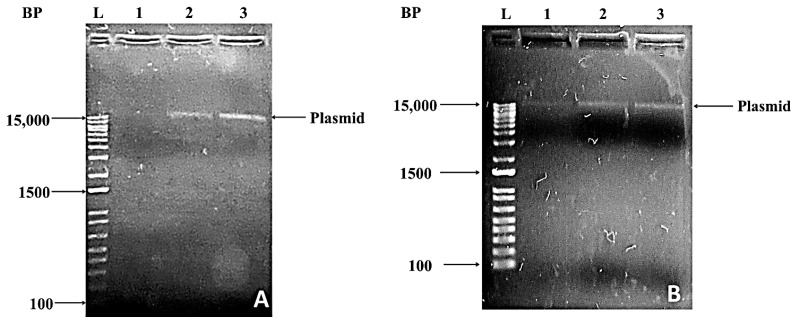
A total of 1.2% agarose gel electrophoresis of plasmid preparation from bacterial isolates BR3 (**A**) and BR4 (**B**). Lanes 1–3 represent the plasmid preparation from culture grown in 1: LB medium, 2: LB medium supplemented with ethidium bromide, and 3: LB medium supplemented with ampicillin. L represents the molecular weight marker; the molecular weight ranges from 100–15,000 bp.

**Figure 5 biotech-11-00004-f005:**
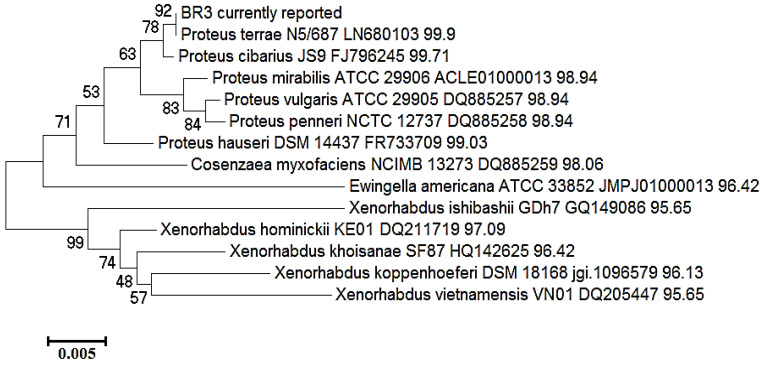
Phylogenetic tree based on 16S-rRNA showing the clustering of BR3 with sequences retrieved form the EzBioCloud database. The tree was constructed in MEGAX by the neighbor-joining method with a 500-bootstrap value.

**Figure 6 biotech-11-00004-f006:**
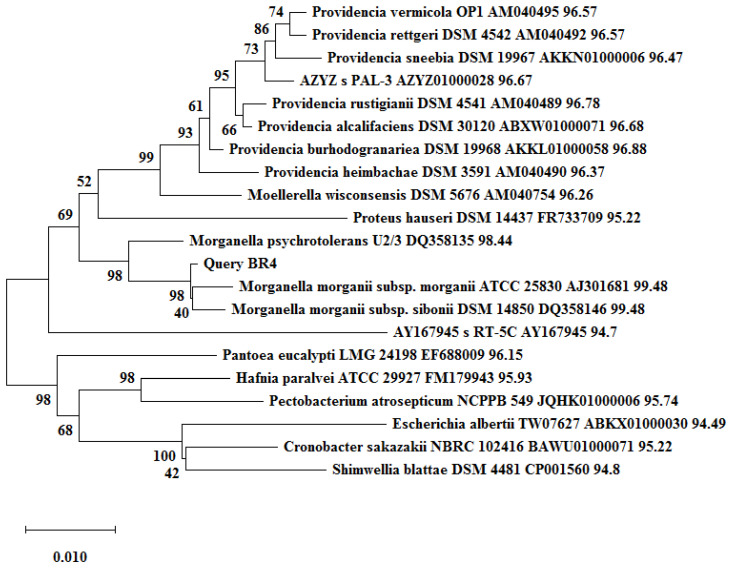
Phylogenetic tree based on 16S-rRNA showing the clustering of BR4 with sequences retrieved form the EzBioCloud database. The tree was constructed in MEGAX by the neighbor-joining method with a 500 bootstrap value.

**Table 1 biotech-11-00004-t001:** Antibiotic sensitivity profile of bacterial isolates BR3 and BR4.

Diameter (mm) of the Zone of Inhibition Following 24 h of Incubation					
Bacterial Isolates	Antibiotics				
	C	V	T	K	G
BR3	32 (S)	11 (R)	11 (R)	23 (S)	22 (S)
BR4	16 (I)	<1 (R)	20 (S)	22 (S)	20 (S)

Letters in parentheses represent: R: resistant, S: sensitive, and I: intermediate. Reference standard inhibition zone sizes: C: chloramphenicol (R- ≤ 12, I-13-17, S- ≥ 18); V: vancomycin (R--, I--, S- ≥ 15); T: tetracycline (R- ≤ 14, I-15-18, S- ≥ 19); K: kanamycin (R- ≤ 13, I-14-17, S- ≥ 18); G: gentamicin (R- ≤ 12, I-13-14, S- ≥ 15). The reference inhibition zone diameter is given by the Antibiotic Sensitivity Teaching kit HTM002-15PR from HiMedia Laboratories Pvt. Ltd., India. Concentration of antibiotics: C: chloramphenicol (30 µg/disc), V: vancomycin (30 µg/disc), T: tetracycline (30 µg/disc), K: kanamycin (30 µg/disc) and G: gentamicin (10 µg/disc).

## Data Availability

Data will be provided on request by the corresponding authors.

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
