# Peer review of "The Identification of Ethidium Bromide-Degrading Bacteria from Laboratory Gel Electrophoresis Waste"

_biotech, 2022, doi:10.3390/biotech11010004_

Round 1

Reviewer 1 Report

The purpose of the article “Identification of ethidium degrading bacteria from laboratory waste” is to reported a characterize of ethidium bromide degrading bacterial isolates for their application and safety point of view including their antibiogram, identification, phylogenetic relationship and pathogenicity. The article, however, must be improved in terms of writing since some grammar and syntax errors are present in the manuscript. They should address the subject and critically review the information from the literature.

 My suggestions:

The authors need to revise the title of the paper in a more meaningful way. “Identification of ethidium bromide degrading bacteria from laboratory gel electrophoresis waste”

The abstract is written in a way lacks logic. It should highlight the salient findings more critically.

Keywords are present in the title, choose others.

Introduction need more convincing rational for this article. 

The introduction has long paragraphs, I suggest reducing the size of the paragraphs.

Provide experimental work plan at the start of M&M. No detail description is available about the experiment.

What statistical method is used?

In the materials and methods, where are described how the Pathogenic Status of BR3 and BR4 analyzes were carried out? Just a bibliographic survey? in my opinion does not fit as results found.

Authors should discuss the results integrally. The discussion is based on individual results. I suggest that integrating the results will give more value to the work. I suggest that you discuss by integrating all your results. You can use correlation tests (PCA or Pearson Correlation). 

The results of this study are not fully explained therefore the interpretation of the results is very difficult. The author needs to provide the % increase or decrease rather than just writing ''significantly increased….''.

Figure 4 and 5, where is the outgroup?

Table 1: Please provide standard error or standard deviation of the results.

Figure 1 and 2: insert figure scales.

Figure 3: insert molecular mass marker and improve the quality of the gel.

The discussion is poorly written hence, needs rewriting. The discussion should be further strengthened by adding some more relevant papers. The literature search is insufficient, only few related research papers in the past three years are cited, add the latest research results appropriately. See the below links if you think it will benefit your discussion.

Rewrite the conclusion! It needs to be much improved.

Author Response

Comment: The purpose of the article “Identification of ethidium degrading bacteria from laboratory waste” is to reported a characterize of ethidium bromide degrading bacterial isolates for their application and safety point of view including their antibiogram, identification, phylogenetic relationship and pathogenicity. The article, however, must be improved in terms of writing since some grammar and syntax errors are present in the manuscript. They should address the subject and critically review the information from the literature.

Response: Thank you for your valuable suggestions. We have incorporated changes throughout the manuscript.

 My suggestions:

Comment: The authors need to revise the title of the paper in a more meaningful way. “Identification of ethidium bromide degrading bacteria from laboratory gel electrophoresis waste”:

Response: Title has been revised.

Comment: The abstract is written in a way lacks logic. It should highlight the salient findings more critically. Response: Thank you for suggestion. Abstract has been revised as per your suggestion.

Comment: Keywords are present in the title, choose others.

Response: Corrected and others keywords have been incorporated.

Comment: Introduction need more convincing rational for this article. 

The introduction has long paragraphs, I suggest reducing the size of the paragraphs.

Response: We have incorporated all your suggestion in the introduction section.

Comment: Provide experimental work plan at the start of M&M. No detail description is available about the experiment.

Response: Work plan has been added as flowchart. Method has either been detailed or reference has been already given. If required all the methods may be detailed further.

Comment: What statistical method is used?

Response: This is qualitative study; therefore, statistics has not been used in this study.

Comment. In the materials and methods, where are described how the Pathogenic Status of BR3 and BR4 analyzes were carried out? Just a bibliographic survey? in my opinion does not fit as results found.

Response: It has been analyzed just based on bibliographic survey after 16srRNA homology based identification of isolates. This has been performed to support the data.

Comment: Authors should discuss the results integrally. The discussion is based on individual results. I suggest that integrating the results will give more value to the work. I suggest that you discuss by integrating all your results. You can use correlation tests (PCA or Pearson Correlation). 

Response: Thank you for your valuable suggestions. We have added/deleted several lines in the discussion section integrating the results. We will improve

Comment: The results of this study are not fully explained therefore the interpretation of the results is very difficult. The author needs to provide the % increase or decrease rather than just writing ''significantly increased….''.

Response: The authors did not find use of ''significantly increased” in this manuscript. This study is based on qualitative measurements.

Comment: Figure 4 and 5, where is the outgroup?

Response: We did online 16S-rRNA sequence similarity-based identification of isolates. Organisms of another genus may be considered as outgroup.

Comment: Table 1: Please provide standard error or standard deviation of the results.

Response: It was screened once. Not applicable.

Comment. Figure 1 and 2: insert figure scales.

Response: Inner diameter of the medium holding plate measures 86 mm.

Comment: Figure 3: insert molecular mass marker and improve the quality of the gel.

Response: Molecular mass marker has been included and quality has been improved.

Comment: The discussion is poorly written hence, needs rewriting. The discussion should be further strengthened by adding some more relevant papers. The literature search is insufficient, only few related research papers in the past three years are cited, add the latest research results appropriately. See the below links if you think it will benefit your discussion.

Response: Thank you for the suggestion. We have added few lines to support the results along with references. We will improve more in proof read of the manuscript.

Comment: Rewrite the conclusion! It needs to be much improved.

Response: Conclusion has been improved in the revised version of the manuscript.

Reviewer 2 Report

The manuscript entitled “Identification of ethidium degrading bacteria from laboratory waste” explored the characterized the EtBr degrading bacterial isolates and the risk of using these isolates for treatment of lab waste in laboratory settings.

 The idea of the study is interesting, but please consider the following comments:

  • Add bromide to the manuscript title to be ethidium bromide.
  • Revise the language of the manuscript in addition to the punctuation.

Author Response

Comment: The manuscript entitled “Identification of ethidium degrading bacteria from laboratory waste” explored the characterized the EtBr degrading bacterial isolates and the risk of using these isolates for treatment of lab waste in laboratory settings.

 The idea of the study is interesting, but please consider the following comments:

  • Add bromide to the manuscript title to be ethidium bromide.
  • Revise the language of the manuscript in addition to the punctuation.

Response: Thank you for your suggestions and feedback on the manuscript. We have incorporated all your suggested changes in the manuscript.

Round 2

Reviewer 1 Report

Thanks for attending the suggestions. The manuscript has been significantly improved. In view of the above, I believe that the article presents robust and consolidated content.